
# Iodide-CIMS and *m/z* 62: The detection of HNO₃ as NO₃⁻ in the presence of PAN, peracetic acid and O₃

Raphael Dörich,[1] Philipp Eger,[1] Jos Lelieveld[1] and John N. Crowley.[1]

[1]Atmospheric Chemistry Department, Max Planck Institute for Chemistry, 55128, Mainz, Germany

*Correspondence to*: John N. Crowley (john.crowley@mpic.de)

**Abstract.** Chemical Ionisation Mass Spectrometry (CIMS) using I⁻ (the iodide anion) as primary reactant ion has previously been used to measure $NO_3$ and $N_2O_5$ both in laboratory and field experiments. We show that reports of the large daytime mixing ratios of $NO_3$ and $N_2O_5$ (usually only present in detectable amounts at night-time) are likely to be heavily biased by the ubiquitous presence of $HNO_3$ in the troposphere and lower stratosphere. We demonstrate in a series of laboratory experiments that the CIMS detection of $HNO_3$ at *m/z* 62 using I⁻ ions is efficient in the presence of PAN or peracetic acid (PAA) and especially $O_3$. We have characterised the dependence of the sensitivity to $HNO_3$ detection on the presence of acetate anions ($CH_3CO_2^-$, *m/z* 59, from either PAN or PAA). The loss of $CH_3CO_2^-$ via conversion to $NO_3^-$ in the presence of $HNO_3$ may represent a significant bias in I-CIMS measurements of PAN and $CH_3C(O)OOH$. The largest sensitivity to $HNO_3$ at *m/z* 62 is achieved in the presence of ambient levels of $O_3$ whereby the thermodynamically disfavoured, direct reaction of I⁻ with $HNO_3$ to form $NO_3^-$ is bypassed by the formation of $IO_X^-$ which react with $HNO_3$ to form e.g. iodic acid and $NO_3^-$. The ozone and humidity dependence of the detection of $HNO_3$ at *m/z* 62 was characterised in laboratory experiments and applied to daytime, airborne measurements in which very good agreement with measurements of the I⁻($HNO_3$) cluster-ion (specific for $HNO_3$ detection) was obtained.

## 1 Introduction

The use of iodide anions (I⁻) as primary ions in mass-spectrometric studies of ion-molecule reactions has a long history. Fehsenfeld et al. (1975) and Davidson et al. (1978) established that the nitrate anion ($NO_3^-$, *m/z* 62) was formed in a rapid reaction between the iodide anion (I⁻) and $N_2O_5$. $NO_3^-$ was also identified as the main product of the reaction between I⁻ with $ClONO_2$ (Huey et al., 1995). The large rate constants for reaction of I⁻ with $N_2O_5$ and $ClONO_2$ led to the development of Chemical Ionisation Mass Spectrometry (CIMS) using I⁻ primary ions (henceforth I-CIMS) in kinetic studies of heterogeneous, atmospheric reactions (e.g. (Hanson and Ravishankara, 1991)) and more recently has found widespread deployment for measurement of atmospheric trace gases ((Huey (2007) and references therein). Early field measurements utilised I-CIMS to detect $N_2O_5$ and peroxyacetyl nitric anhydride (PAN, $CH_3C(O)O_2NO_2$) (Slusher et al., 2004) but since then the range of molecules that have been detected using I⁻ has greatly increased and trace-gases as diverse as inorganic radicals and halogenates and a host of organic species are now routinely measured (Huey, 2007; Lee et al., 2014; Iyer et al.,



2017; Riva et al., 2019). In this work, we focus on the detection of two atmospherically important trace gases $N_2O_5$ and $HNO_3$ using a CIMS operating with $I^-$ reactant ions.

Both $N_2O_5$ and $HNO_3$ are formed in the atmosphere by the sequential oxidation of NO, which has both anthropogenic and natural sources. In a very well-known series of reactions, NO is oxidised (R1, R2) by reaction with $O_3$ or peroxyl radicals

($RO_2$) to $NO_2$, which during the day, may be removed by reaction with OH to form $HNO_3$ (R3) and during the night to form $N_2O_5$ (R4, R5).

| | | | |
|---|---|---|---|
| $NO + O_3$ | $\rightarrow$ | $NO_2 + O_2$ | (R1) |
| $NO + RO_2$ | $\rightarrow$ | $NO_2 + O_2$ | (R2) |
| $NO_2 + OH + M$ | $\rightarrow$ | $HNO_3 + M$ | (R3) |
| 40   $NO_2 + O_3$ | $\rightarrow$ | $NO_3 + O_2$ | (R4) |
| $NO_3 + NO_2 + M$ | $\rightarrow$ | $N_2O_5 + M$ | (R5) |

Both $HNO_3$ and $N_2O_5$ have important, non-gas loss processes such as uptake to particles and other surfaces. In addition, $N_2O_5$ can thermally dissociate back to $NO_3$.

The chain of reactions to form $N_2O_5$ is broken during the day as $NO_3$ is rapidly photolysed and also reacts with NO so that

$N_2O_5$ is expected to be present at significant levels only at night-time.

The detection of $N_2O_5$ using $I^-$ reactant ions can be achieved by monitoring either the $NO_3^-$ product at $m/z$ 62 (see above) or the adduct ion at $m/z$ 235 (Kercher et al., 2009). The former is reported to be more sensitive and less dependent on water vapour concentrations but less specific, with large and highly variable background signals potentially arising from trace gases such as $NO_3$, $ClONO_2$ and $BrONO_2$. Despite this, night-time $N_2O_5$ has been monitored in ambient air (as $NO_3^-$) using $I^-$

reactant ions, showing reasonable agreement with optical methods (Slusher et al., 2004; Dubé et al., 2006; Chang et al., 2011).

During a recent, airborne deployment of our I-CIMS, we monitored $NO_3^-$ at $m/z$ 62 in an attempt to detect $N_2O_5$ during two night-time flights. The air masses we investigated were mainly in the tropical free and upper troposphere and lower stratosphere and we did not expect significant interference from e.g. halogen nitrates at $m/z$ 62. However, our airborne

measurements (described in detail in section 4) revealed a large and variable signal at $m/z$ 62 both during the day and night. To illustrate this, raw signals obtained during daytime when the aircraft sampled air masses with varying degrees of stratospheric influence are displayed in Fig. S1. The signal at $m/z$ 62 is large and highly variable and is not affected by addition of NO to the heated inlet, ruling out its assignment to either $N_2O_5$ or $NO_3$ (see below). The great increase in signal when entering the lower stratosphere and the obvious correlation with $O_3$ (Fischer et al., 1997; Popp et al., 2009) provided an

early clue to the identity of the trace-gas detected at $m/z$ 62 which we initially assigned to $HNO_3$. Our results thus appeared to contrast the conclusions of a previous observation of a large daytime signal at $m/z$ 62 when deploying an I-CIMS (in this case in the boundary layer), which was interpreted as resulting (at least in part) from high levels of daytime $NO_3$ and/or $N_2O_5$





(Wang et al., 2014). Based on complementary laboratory experiments, Wang et al. (2014) showed, in accord with earlier investigations (Fehsenfeld et al., 1975; Huey et al., 1995), that $HNO_3$ is not detected sensitively at $m/z$ 62 using I-CIMS.

The unexpected observation of a large daytime signal at $m/z$ 62 during airborne operation led us to perform a series of laboratory experiments to identify potential "interfering" trace gases at this mass-to-charge ratio when using I-CIMS. In contrast to the conclusions drawn from previous studies, our laboratory and airborne measurements conclusively show that, during daytime, the predominant contributor to $m/z$ 62 when sampling ambient air (in the presence of ozone) is likely to be $HNO_3$.

## 2 Experimental details

The I-CIMS we used in our laboratory and airborne investigations (see Fig. 1) is similar to that described by Slusher et al. (2004) and Zheng et al. (2011) and was originally constructed in collaboration with Georgia Tech as a prototype instrument of the company THS (http://thsinstruments.com). It is essentially a hybrid of the instruments described by Phillips et al. (2013) and Eger et al. (2019), the former using a $^{210}Po$ ion source, the latter an electrical discharge source but with improved

(digital) control of the MS settings enabling different mass-to-charge ratios to be monitored using different potentials for the collisional dissociation of cluster ions. For all the experiments described below, the $^{210}Po$ ion source was used to generate $I^-$ as this configuration has much better sensitivity for PAN, the main target trace gas during the deployment of the I-CIMS on the HALO aircraft (High Altitude Long range platform for atmospheric Observations). The set-up for PAN detection includes a heated inlet section (~170 °C, 100 mbar, residence time ~ 40 ms) to thermally dissociate PAN to $CH_3C(O)O_2$

which subsequently reacts with $I^-$ to form the acetate anion ($CH_3CO_2^-$) which is detected at $m/z$ 59. At this inlet temperature and pressure, the thermal decomposition rate constant for PAN is 380 $s^{-1}$ implying a lifetime of ~2 ms. For $N_2O_5$, the rate coefficient for its thermal dissociation to $NO_2$ and $NO_3$ is 1940 $s^{-1}$ (lifetime of ~0.5 ms) so that $N_2O_5$ is stoichiometrically converted to $NO_3$ and the instrument measures the sum of $N_2O_5$ and $NO_3$ at $m/z$ 62. In order to separate PAN signals from those of peracetic acid ($CH_3C(O)OOH$, also detected as $CH_3CO_2^-$ at $m/z$ 59) we periodically add NO to the inlet to remove

$CH_3C(O)O_2$ and thus eliminate sensitivity to PAN. As NO reacts more rapidly with $NO_3$ than with $CH_3C(O)O_2$ at 170 °C ($k_{NO+NO3}$ = 2.3 × $10^{-11}$ $cm^3$ molecule$^{-1}$ $s^{-1}$ , $k_{NO+CH3C(O)O2}$ = 1.4 × $10^{-11}$ $cm^3$ molecule$^{-1}$ $s^{-1}$) the concentration of NO added is also sufficient to quantitatively titrate $NO_3$ to $NO_2$ and thus provides a measure of the "background" signal at $m/z$ 62 in the absence of $NO_3$ and $N_2O_5$.

During airborne operation on HALO, the dynamic pressure generated in a forward facing trace gas inlet (TGI) located on top

of the aircraft (see Fig. 1) was used to create a flow of air through ¼ inch (OD) PFA tubing sampling at an angle of 90° to the flight direction. The ¼ inch tubing was atached to a ½ inch (OD) PFA tube attached to an exhaust plate at the underside of the aircraft to create a fast "bypass" flow. The bypass flow was sub-sampled (again at 90° and by ¼ inch PFA tubing heated to 40°C) by the 1.4 L (STP) min$^{-1}$ flow into the I-CIMS. Sub-sampling twice at 90° to the flow was helpful in reducing the number of large particles (e.g. cloud droplets) that could enter the thermal dissociation inlet and IMR.



The thermal dissociation inlet of the I-CIMS is regulated to a pressure of 100 mbar, which results in a pressure in the ion-molecule reactor of 24 mbar. This way, a stable pressure in the thermal dissociation inlet and the Ion Molecule Reactor (IMR) was maintained at altitudes up to ~15 km. Prior to take off, the inlet line and TGI were flushed with nitrogen to prevent contamination by the high levels of pollutant trace gases at the airport. As described in Eger et al. (2019) negative ions exiting the IMR were declustered in passage through a collisional dissociation region (CDC, 0.6 mbar) before passing

through an octopole ion-guide ($6 \times 10^{-3}$ mbar) and a quadrupole for mass selection ($9 \times 10^{-5}$ mbar) prior to detection using a channeltron.

I⁻ ions were generated by combining flows of 4 cm³ (STP) min⁻¹ $CH_3I/N_2$ (400 ppmv) with 750 cm³ (STP) min⁻¹ $N_2$ and passing the mixture through a 370 MBq $^{210}Po$ source. Under standard operating conditions (including airborne deployment), a constant amount of $H_2O$ was added to the IMR by flowing 50 sccm $N_2$ (at 1 bar pressure) through a 30 cm length of water-

permeable 1/8-inch tubing (Permapure) immersed in water. The 50 sccm flow of $N_2$ acquires a relative humidity close to 100 % in transit through the tubing and is subsequently mixed with the dry $N_2$ flow and sample air. Under these conditions, the ratio of signals at $m/z$ 145 (I⁻($H_2O$)) to that at $m/z$ 127 (I⁻) was 0.068. By comparison with calibration curves (see Fig S2 and associated text) this indicates an $H_2O$ concentration in the IMR of ~$4 \times 10^{14}$ molecule cm⁻³. For laboratory tests, the amount of water in the IMR could be increased by reducing the pressure in the permeable tube (thus increasing the mole fraction of

$H_2O$) or set to zero by bypassing the humidifier.

Based on a (calculated) literature value for the free energy of formation of I⁻($H_2O$)₁ of -6.1 kcal mol⁻¹ (Teiwes et al., 2019) we derive an equilibrium constant (at 298 K) of $K_6 = 1.16 \times 10^{-15}$ cm³ molecule⁻¹ for the formation and thermal dissociation of I⁻($H_2O$)₁

I⁻ + $H_2O$          ⇌          I⁻($H_2O$)₁                                                          (R6)

With an $H_2O$ concentration (in the IMR) of $3.9 \times 10^{14}$ molecule cm⁻³ this implies that the ratio [I⁻($H_2O$)₁ ] / [I⁻] = 0.45. Our measured ratio of signals at $m/z$ 145 (I⁻($H_2O$)) / $m/z$ 127 (I⁻) was a factor ~6 lower, reflecting the fact that, even when the declustering potential is reduced to its minimum value, most I⁻($H_2O$) ions do not survive the CDC region.

During extended operation of the CIMS, changes in sensitivity were captured by monitoring the primary ion signal (I⁻ and its water cluster). Background signals at each of the mass-to-charge ratios monitored were obtained by passing the sampled air

through a tubular scrubber (alluminium) filled with stainless-steel wool heated to 120 °C.

## 3 Laboratory Characterisation

As described above, our observations of a clear correlation between $m/z$ 62 and $O_3$ mixing ratios during the first HALO deployment of the I-CIMS strongly suggested that $HNO_3$ was the origin of the signal although previous experiments had shown that I⁻ does not react with $HNO_3$ to form $NO_3^-$. In order to determine the sensitivity of our I-CIMS to $HNO_3$ we

constructed a permeation source in which a 20 cm³ (STP) min⁻¹ flow of zero air was passed through a 1m length of PFA tubing (0.125 inch OD) which was formed into a coil and submerged in an aqueous solution of 65% $HNO_3$ held at 50°C. The





permeation rate was determined by passing the 20 cm$^3$ (STP) min$^{-1}$ flow through an optical absorption cell and measuring the optical extinction at 185 nm where the absorption cross-section of HNO$_3$ is well known (Dulitz et al., 2018). For the I-CIMS calibration, the 20 cm$^3$ (STP) min$^{-1}$ output was dynamically diluted to generate a mixing ratio of between 5 and 50 ppbv.

Based on uncertainties in the absorption cross-section (5%), the reproducibility of the optical measurement and the dilution factor, the uncertainty of the HNO$_3$ mixing ratio is estimated as 15 %.

Figure 2 shows the response of the I-CIMS at $m/z$ 62 to addition of various amounts of HNO$_3$. Throughout the paper when presenting raw data, we normalise the I-CIMS signal by dividing by the primary ion signal at $m/z$ 127. The weak signal in the absence of O$_3$ (blue data points) confirms the conclusions of previous studies that derive a low rate coefficient for

reaction (R6). For comparison, approximate, relative sensitivities to PAN ($m/z$ 59), N$_2$O$_5$ ($m/z$ 62) and HNO$_3$ ($m/z$ 62), using this instrument are 1, 0.1 and $5 \times 10^{-4}$, respectively. Indeed, as written below, reaction (R7) is endothermic by ~43 kJ mol$^{-1}$ (Goos et al., 2005).

$$I^- + HNO_3 \quad \rightarrow \quad NO_3^- + HI \quad\quad\quad\quad\quad\quad (R7)$$

In a further series of experiments, we measured the response of the I-CIMS to HNO$_3$ when adding O$_3$ to the zero-air. The

results, also plotted in Fig. 2 (black symbols), indicate a factor ~250 increase in the signal at $m/z$ 62 when ~500 ppbv ozone was added. There are two possible explanations for this observation. The first involves conversion of NO$_2$ impurity (that is present as a ~8 % impurity in the HNO$_3$ permeation flow) to NO$_3$ and N$_2$O$_5$ (R1, R4, R5) which are subsequently detected. This can however be ruled out as the rate-limiting step in the formation of NO$_3$ is the slow reaction between NO$_2$ and O$_3$ with $k_4 = 3.5 \times 10^{-17}$ cm$^3$ molecule$^{-1}$ s$^{-1}$ at room temperature (Atkinson et al., 2004). The addition of 1000 ppbv O$_3$

(equivalent to a concentration of $2.4 \times 10^{13}$ molecule cm$^{-3}$) would only convert an insignificant fraction of the NO$_2$ to NO$_3$ in the ~40 ms reaction time available from the point of mixing to the IMR. This could be confirmed by adding NO (7.7 ppm) to the inlet which would remove any NO$_3$ (see above) and observing no change in the signal at $m/z$ 62.

The second explanation is that the presence of O$_3$ results in the generation of further reagent ions that can react with HNO$_3$. Iodide anions are known to react with O$_3$, leading, in a series of exothermic reactions, to the formation of iodate (Williams et

al., 2002; Teiwes et al., 2018; Bhujel et al., 2020).

$$I^- + O_3 \quad \rightarrow \quad IO^- + O_2 \quad\quad\quad\quad\quad\quad (R8)$$

$$IO^- + O_3 \quad \rightarrow \quad IO_2^- + O_2 \quad\quad\quad\quad\quad\quad (R9)$$

$$IO_2^- + O_3 \quad \rightarrow \quad IO_2^- + O_2 \quad\quad\quad\quad\quad\quad (R10)$$

In this scheme, R8 is rate-limiting ($k_8$ ~$1 \times 10^{-12}$ cm$^3$ molecule$^{-1}$ s$^{-1}$ (Bhujel et al., 2020), whereas the further steps (R9-R10)

in the sequential oxidation to iodate proceed with rate constants at least two orders of magnitude larger (Teiwes et al., 2018; Bhujel et al., 2020). IO$^-$ and IO$_2^-$ also react with O$_2$ to reform O$_3$:

$$IO^- + O_2 \quad \rightarrow \quad I^- + O_3 \quad\quad\quad\quad\quad\quad (R11)$$

$$IO_2^- + O_2 \quad \rightarrow \quad IO^- + O_3 \quad\quad\quad\quad\quad\quad (R12)$$





with rate coefficients of $k_{11} = 3.2 \times 10^{-14}$ cm$^3$ molecule$^{-1}$ s$^{-1}$ and $k_{12} = 1.3 \times 10^{-14}$ cm$^3$ molecule$^{-1}$ s$^{-1}$ (Bhujel et al., 2020). With
the O$_3$ concentrations (~1-5 × 10$^{10}$ molecule cm$^{-3}$) and reaction times used in these studies (Teiwes et al., 2018; Teiwes et al., 2019; Bhujel et al., 2020) IO$_3^-$ was observed to be the dominant form of IO$_X^-$.

In the presence of water vapour, I$^-$ is also present as a hydrate I$^-$(H$_2$O) (see above) for which, according to Teiwes et al. (2019), the rate coefficient for reaction with O$_3$ (R13a, R13b) is a factor ~40 larger than $k_8$ and results in the formation of IO$_2^-$ and I$^-$:

I$^-$(H$_2$O) + O$_3$      →       IO$_2^-$ + neutrals                                                        (R13a)

I$^-$(H$_2$O) + O$_3$      →       I$^-$ + neutrals                                                          (R13b)

As R8 is rate limiting, this implies an increase in the amount of e.g. IO$_3^-$ formed in the IMR in the presence of water. In most regions of the troposphere and lower atmosphere ozone mixing ratios lie between 30 and >1000 ppbv. An ambient ozone concentration of 50 ppbv results in a concentration in the IMR of > 10$^{10}$ molecules cm$^{-3}$. The large rate coefficients for R9
and R10 and the reactions of IO$^-$ and IO$_2^-$ with O$_2$ result in the rapid inter-conversion of I$^-$, IO$^-$, IO$_2^-$ and IO$_3^-$ which results (for a given RH and ozone concentration) in a quasi-equilibrium between IO$_X^-$ anions.

We explored the relevance of these reactions for our I-CIMS by carrying out a set of experiments in which varying amounts of O$_3$ were added to the inlet and the mass-to-charge ratios corresponding to IO$^-$ (m/z 143), IO$_2^-$ (m/z 159) and IO$_3^-$ (m/z 175) were monitored; the results are depicted in Fig. 3.

First, we note that all three mass-to-charge ratios were indeed observed, but only under conditions where the CDC potential was set to the lowest value at which ions still reach the detector. The dependence of the various IO$_X^-$ anions on the O$_3$ mixing ratio is broadly as expected from the reaction scheme (R7-R11) listed above: The major contributor to IO$_X^-$ at low [O$_3$] is IO$^-$, which is converted to IO$_2^-$ and IO$_3^-$ more efficiently as O$_3$ increases, while the total concentration of IO$_X^-$ increases approximately linearly. At the maximum O$_3$ mixing ratio used (577 ppbv) there are (following dilution) 375 ppbv in the
IMR, which translates to a concentration (at 24 mbar and ~298 K) of $2.1 \times 10^{11}$ molecule cm$^{-3}$. This O$_3$ concentration is comparable with those used by Teiwes et al. (2018) (~1-4 10$^{11}$ molecule cm$^{-3}$) or Bhujel et al. (2020) (~4 × 10$^{10}$ molecule cm$^{-3}$) in their ion-trap based, kinetic investigations of the formation of IO$_X^-$ when reacting I$^-$ with O$_3$. Their observation that IO$_3^-$ is the dominant anion is however not consistent with our results, which indicate that IO$_3^-$ represents only ~35% of the total IO$_X^-$ signal. The relative abundance of each IO$_X^-$ depends not only on the O$_3$ concentration but also on the reaction
time, which, for both Teiwes et al. (2018) and Bhujel et al. (2020) was between 10-100 ms. Based on the flow into the IMR, its volume (~50 cm$^3$) and the pressure we calculate a similar residence time (for neutrals) of about 25 ms. We also considered the possibility that the application in our I-CIMS of a potential difference between the entrance and exit of the IMR (to optimise ion-transmission) could result in a significantly shorter IMR-residence time for ions. This was assessed by calculating the drift-velocity ($V_d$) in the IMR from the electric field strength ($E$ ~12 Vm$^{-1}$) and the ion mobility (μ).

$V_d = E\mu$                                                                                          (1)



The electrical mobility of $I^-$ was calculated for our conditions (using the Mason-Schamp equation) as ~ 0.15 $m^2 V^{-1} s^{-1}$ using a collision cross-section (for an $I^- / N_2$ pair) of $9 \times 10^{-16}$ $cm^2$ molecule$^{-1}$ (McCracken, 1952). Via equation (1), this results in a drift velocity of 1.8 m s$^{-1}$, or an ion residence time (in the ~8 cm long IMR) of 44 ms, which is comparable to the residence time of neutrals. Our observation that $IO^-$ (and not $IO_3^-$) is the dominant ion-signal is thus unlikely to result from differences

in reaction times, temperature (our IMR is at ~15°C above ambient temperature owing to the heated inlet) or $O_3$ concentrations in the different set-ups but may be related to the higher pressure of $O_2$ (> 2 mbar) in our IMR which converts $IO_3^-$ back to $IO^-$ thus competing with further oxidation (via reaction with $O_3$) to $IO_3^-$. Additionally, the high IMR pressures (24 mbar) in our experiments are ~ six orders of magnitude higher than the ~$10^{-5}$ mbar available in the ion-trap experiments of Teiwes et al. (2018) and Bhujel et al. (2020) which will result in more rapid thermalization of the ions present and prevent

potentially non-thermal reactions and thus bias to the rate coefficients derived.

The effect of adding $H_2O$ to the IMR was explored in a further set of experiments and the variation of the signals at mass-to-charge ratios corresponding to $IO_X^-$ with [$H_2O$] are displayed in Fig. 4. The experiments were carried out with the $O_3$ mixing ratio fixed at either 70 or 120 ppbv, close to that typically found in the lower troposphere (~20-100 ppbv). At the lowest $H_2O$ concentrations in our experiments, the total $IO_X^-$ signal is about 30 counts. This increases by a factor of ~10 when [$H_2O$]$_{IMR}$

increases to $3 \times 10^{15}$ molecule cm$^{-3}$. Increasing the $O_3$ mixing ratio from 70 to 120 ppbv results in an increase in the signals at $m/z$ 175 ($IO_3^-$) and $m/z$ 159 ($IO_2^-$) at all water vapour concentrations, whereas the signal at $m/z$ 143 ($IO^-$) is reduced at the lowest water vapour concentrations. These observations reinforce the concept of a larger rate coefficient for reaction of $I^-(H_2O)$ with $O_3$ (R12a) compared to $I^-$ (R7) (Teiwes et al., 2019) and the sequential conversion of $IO^-$ to more oxidized forms as described by equation R7-R9.

Having established that all of the expected $IO_X^-$ anions are present in our IMR, we can propose a route for $HNO_3$ detection as $NO_3^-$ which involves transfer of a proton from $HNO_3$ (a very strong acid) to the conjugate base of the respective iodine containing acids (hypoiodous-, iodous- and iodic-acid):

$IO^-$ + $HNO_3$ → $NO_3^-$ + HOI (R13)

$IO_2^-$ + $HNO_3$ → $NO_3^-$ + HOIO (R14)

$IO_3^-$ + $HNO_3$ → $NO_3^-$ + $HOIO_2$ (R15)

Taking $IO_3^-$ as an example, we see that the net reaction, ($I^-$ + $O_3$ + $HNO_3$ → $NO_3^-$ + $HOIO_2$) is driven by the relative stability of iodic acid compared to $O_3$, thus bypassing the thermodynamic barrier to direct formation of $NO_3^-$ from $HNO_3$ + $I^-$. As described above, the $O_3$ dependence of the ion signals we observe for $IO^-$, $IO_2^-$ and $IO_3^-$ are consistent with the sequential oxidation of $I^-$ by $O_3$. However, the relative ion-abundance we observe at the detector does not necessarily reflect the relative

concentration of the ions in the IMR and we cannot assign the individual contribution of any single $IO_X^-$ anion to $HNO_3$ detection. We are unable to completely shut of collisional dissociation in our I-CIMS which may be a characteristic that is peculiar to our instrument as we do not detect weakly-bound $I^-(R(O)OH)$ clusters which are commonly monitored in other





instruments utilising $I^-$ chemical ionisation (Lee et al., 2014). Hence, our relative sensitivity to the $IO_X^-$ components is unknown.

In order to confirm that $IO_X^-$ is responsible for detection of $HNO_3$ we examined the depletion of the signals at $m/z$ 143, $m/z$ 159 and $m/z$ 175 when adding very large concentrations of $HNO_3$ to the IMR. The results, summarised in Fig. 5, indicate that all three $IO_X^-$ ions are removed when the $HNO_3$ mixing ratio was increased from zero to 80 ppbv, but with different fractional changes. This can be understood if e.g. the individual $IO_X^-$ react with $HNO_3$ with different rate coefficients. The solid lines in Fig. 5 represent exponential decays of each ion, with rate coefficients of $\sim 10 \times 10^{-10}$ cm$^3$ molecule$^{-1}$ s$^{-1}$ for

$HNO_3 + IO_3^-$, $\sim 7 \times 10^{-10}$ cm$^3$ molecule$^{-1}$ s$^{-1}$ for $HNO_3 + IO_2^-$ and $\sim 3 \times 10^{-10}$ cm$^3$ molecule$^{-1}$ s$^{-1}$ for $HNO_3 + IO^-$. These approximate values were derived by converting the $HNO_3$ mixing ratio into a concentration in the IMR and assuming pseudo-first-order behaviour (i.e. negligible depletion of $HNO_3$) so that (using $IO_3^-$ as example):

$S(IO_3^-)_t = S(IO_3^-)_0 \exp(-kt[HNO_3]_{IMR})$

Where $S(IO_3^-)_t$ and $S(IO_3^-)_0$ are the signals at $m/z$ 175 after and prior to addition of $HNO_3$, respectively. $[HNO_3]_{IMR}$ is the

concentration (molecule cm$^{-3}$) of $HNO_3$ in the IMR, $k$ (cm$^3$ molecule$^{-1}$ s$^{-1}$) is the rate coefficient for reaction between $HNO_3$ and $IO_3^-$ and $t$ is the reaction time, which we assume to be 25 ms (see above). This analysis assumes that the re-establishment of equilibria between $IO_X^-$ is minimal on the time scale of the reaction between any single $IO_X^-$ and $HNO_3$. The results indicate qualitatively that $IO_3^-$ is the most reactive of the $IO_X^-$ anions towards $HNO_3$, but that all three contribute to $HNO_3$ detection. The depletion of the summed $IO_X^-$ signals versus the accompanying increase in signal due to $NO_3^-$ at $m/z$ 62 is

displayed in Fig. 6, which indicates a roughly linear relationship, confirming that $IO_X^-$ are mainly responsible for detection of $HNO_3$ in our I-CIMS. We note that the increase in signal at $m/z$ 62 is about a factor 100 greater than the reduction in signal from $IO_X^-$, confirming that the detection of $IO_3^-$ in our instrument is inefficient.

While the reactions of $IO_X^-$ with $HNO_3$ represent the most likely route to $HNO_3$ detection at $m/z$ 62 in our CIMS other possibilities are the reactions of oxide, superoxide and ozone anions ($O_X^-$) and hydrated $O_X^-$ with $HNO_3$ as they have large

rate coefficients ($> 10^{-9}$ cm$^3$ molecule$^{-1}$ s$^{-1}$) and form $NO_3^-$ (Huey, 1996; Wincel et al., 1996; Lengyel et al., 2020):

$O^- + HNO_3$   $\rightarrow$   $NO_3^- + OH$                        (R16)

$O_2^- + HNO_3$   $\rightarrow$   $NO_3^- + HO_2$                        (R17)

$O_3^- + HNO_3$   $\rightarrow$   $NO_3^- + neutrals$                      (R18)

However, when adding $O_3$ (up to 600 ppbv) to the IMR we saw no signal that could be attributable to any oxide anion $O_X^-$.

Figure 7a displays the dependence of the $NO_3^-$ signal at $m/z$ 62 on the $O_3$ mixing ratio with $HNO_3$ held constant at 40 ($\pm$ 6) ppbv. The dependence of the normalised signal at $m/z$ 62 on $[O_3]$ is clearly non-linear. We showed above that the sum of all $IO_X^-$ anions increases approximately linearly with $O_3$ mixing ratio while at the same time the behaviour of $IO^-$ and $IO_3^-$ is more complex. The sensitivity of $HNO_3$ detection to increases in $O_3$ is expected to depend not only on the individual concentrations of each anion in the IMR but also on their respective rate coefficients for reaction with $HNO_3$ (which are





different, see above) and the observed non-linearity is not surprising. The solid lines through the data points are regressions
of the form:

$$\text{Signal } (m/z\ 62) = A(1-\exp(-[O_3]B)) \tag{2}$$

Where $[O_3]$ is the $O_3$ mixing ratio in ppbv and $B$ has a value of $1.515 \times 10^{-3}$ per ppbv of $O_3$. As shown in Fig 7b, for a given
$[HNO_3]$, the parameter $A$ is dependent on the water vapour concentration (i.e. on ratio of signals at $m/z$ 145 and $m/z$ 127,
$(S_{145} / S_{127})$ over the range explored and can be parameterised as:

$$A = 0.138 + 0.929 \times (S_{145} / S_{127}) \tag{3}$$

In these experiments, $H_2O$ was not added to the TD inlet (this would have increased the retention time of $HNO_3$ on inlet
surfaces, thereby making changes in the $m/z$ 62 difficult to interpret) but directly to the IMR, as described in section 2 and as
used during airborne operation of the CIMS. The linear dependence of the signal at $m/z$ 62 on the ratio of signals at $m/z$ 145
and $m/z$ 127 at various ozone concentrations ($[HNO_3]$ fixed at 38.5 ppbv is further highlighted in Fig S3.

The positive intercept in Fig. 7b indicates that there is significant sensitivity to $HNO_3$ detection at $m/z$ 62 in the absence of
water in the IMR, implying that $IO_X^-$ anions can react directly with $HNO_3$ to form $NO_3^-$ as written in R13-15. The increase in
the sensitivity to $HNO_3$ as the water vapour concentration is increased is consistent with the formation of $I^-(H_2O)$ ($m/z$ 145)
which reacts more rapidly with $O_3$ (to form $IO_2^-$ directly) than does $I^-$ (Teiwes et al., 2019), thereby increasing the abundance
of $IO_X^-$ in the IMR (see above) and thus the instrument's sensitivity to $HNO_3$.

The very strong sensitizing effect of ozone and $H_2O$ vapour can explain why similar instruments to ours observe large
signals at $m/z$ 62 when sampling ambient air. Indeed, both $O_3$ and $HNO_3$ are ubiquitous and generally present at much high
levels than either $NO_3$ or $N_2O_5$ and attempts to measure these traces gases using I-CIMS without TD-inlets and NO titration
(to remove the $HNO_3$ contribution) will likely result in erroneously high levels of both, especially during the day when
lower-tropospheric $O_3$ and $HNO_3$ are often at their highest levels. It also explains why laboratory tests (generally carried out
without added $O_3$ or $H_2O$) have shown only low (or no) sensitivity to $HNO_3$ at $m/z$ 62.

We have also evaluated the potential for "unintentional" $HNO_3$ detection at $m/z$ 62 by its reaction with the acetate anion,
$CH_3CO_2^-$:

$$CH_3CO_2^- + HNO_3 \rightarrow NO_3^- + CH_3C(O)OH \tag{R19}$$

The $CH_3CO_2^-$ anion is the conjugate base of a weak acid ($CH_3C(O)OH$) has been utilised to monitor a number of trace gases
via proton transfer (Veres et al., 2008). While Veres et al. (2010) generated $CH_3CO_2^-$ deliberately by passing acetic-
anhydride through their $^{210}$Po-source, in our experiments it is the product (monitored at $m/z$ 59) of the reaction between $I^-$
primary ions and either $CH_3C(O)O_2$ (from the thermal dissociation of PAN) or $CH_3C(O)OOH$.

$$I^- + CH_3C(O)O_2 \rightarrow CH_3CO_2^- + IO \tag{R20}$$
$$I^- + CH_3C(O)OOH \rightarrow CH_3CO_2^- + HOI \tag{R21}$$

Figure 8a shows the result of a set of experiments demonstrating $HNO_3$ detection at $m/z$ 62 without (blue data points) and
with 3.25 ppbv of $CH_3C(O)OOH$ (black data points) added to the inlet flow. The initial (normalised) signal at $m/z$ 59 from



the $CH_3CO_2^-$ anion in the absence of $HNO_3$ was 53500 counts. The presence of 3.25 ppbv $CH_3C(O)OOH$ (and resultant $CH_3CO_2^-$) results in a ~50-fold increase in the sensitivity of the I-CIMS to $HNO_3$. We also carried out a few experiments
(less systematic) in which PAN (instead of PAA) was added to the IMR and obtained the same results.

Our results disagree with the conclusions of Wang et al. (2014) who saw no increase at *m/z* 62 when adding PAN to air containing $HNO_3$ but are consistent with the use of $CH_3CO_2^-$ as primary reactant ion to detect $HNO_3$ at *m/z* 62 (Veres et al., 2008). Figure 8b indicates that the increase in signal at *m/z* 62 when adding $HNO_3$ to a flow of $CH_3C(O)OOH$ in air is approximately proportional to the reduction in the ion-signal at *m/z* 59. This helps confirm that $CH_3CO_2^-$ is the ion
responsible for the detection of $HNO_3$ but also indicates that the detection of PAN and $CH_3C(O)OOH$ via conversion to $CH_3CO_2^-$ can be compromised when $HNO_3$ is present in the air sample. Indeed, in many air masses the concentration of $HNO_3$ can be an order of magnitude greater than that of either PAN or $CH_3C(O)OOH$ and given that other abundant trace gases (e.g. organic acids) also react with $CH_3CO_2^-$ (Veres et al., 2008) further reactions of $CH_3CO_2^-$ in the ion-molecule reactor regions of I-CIMS instruments may result in a significant bias (to lower values) which would have to be analysed
case-by-case for different instruments.

Wang et al. (2014) observed that the majority of the *m/z* 62 signal during the daytime could be removed by addition of NO (0.54 ppmv or $1.3 \times 10^{13}$ molecule cm$^{-3}$) to the inlet. At their inlet temperature of 120-180 °C, NO reacts with $O_3$ with a rate coefficient in the range 6-9 $\times 10^{-14}$ cm$^3$ molecule$^{-1}$ s$^{-1}$, which results in a half-life for $O_3$ of 500 to 800 ms. (Wang et al., 2014) do not mention the residence time of air passing through their heated inlet, but it appears plausible that a substantial fraction
of ambient $O_3$ would have been removed during background measurement, thus decreasing (or removing) sensitivity towards $HNO_3$ via reactions involving $O_3$ in the IMR, and leading the authors to conclude that $NO_3$ was being detected above a lower background than truly present.

To illustrate the potential size of the bias due to $HNO_3$ when monitoring $N_2O_5$ at *m/z* 62 in field measurements we take the relative sensitivities (at *m/z* 62) of our I-CIMS to $N_2O_5$ and to $HNO_3$ (in the presence of typical boundary layer mixing ratios
of $O_3$ (50 ppbv) and at typical relative humidity (50%). Under these conditions, with $N_2O_5$ and $HNO_3$ mixing ratios of 0.2 and 2 ppbv, respectively, we calculate that $HNO_3$ would account for > 70% of the signal at *m/z* 62.

## 4 Field Measurements

Having shown that $HNO_3$ is detected by our I-CIMS with reasonable sensitivity when sufficient $O_3$ is present in ambient air samples, we now examine the signals at *m/z* 62 obtained in airborne operation of the I-CIMS during two CAFE (Chemistry
of the Atmosphere Field Experiment) campaigns of the HALO aircraft. In the CAFE-Africa campaign (2018) the I-CIMS monitored *m/z* 62 on several flights over the Atlantic west of the African continent. During the 2020 CAFE-EU campaign with HALO over Europe, the I-CIMS additionally monitored *m/z* 190 (the $I^-(HNO_3)$ cluster ion) which is selective to $HNO_3$.



## 4.1 CAFÉ-Africa

Here we examine the results obtained during a HALO flight as part of the CAFE-Africa mission. The flight in question was
the transfer from Sal airport on the Cape Verde islands (which served as base-station during the mission) back to Germany.
During the flight the aircraft flew mainly at high altitudes (13-15 km) so that stratospheric air was sampled at higher latitudes
but also made two dives into the free-troposphere. The flight track is displayed in Fig S4.

Figure 9 shows a time-series of ozone mixing ratios during the flight (in red, panel a) along with the I-CIMS signal at $m/z$ 62
(in red, panel b). In air masses with stratospheric influence (i.e. $O_3$ values > 100 ppb, 12:20 -15:10 UTC) there is an obvious,
strong co-variance between these two parameters. However, once corrected for the dependence of the sensitivity of the I-
CIMS to $O_3$ (equations 2 and 3) we obtain the black line representing the mixing ratios of $HNO_3$ and the covariance is
greatly reduced. We also note that, apart from some significant increases at ~11:30 and ~16:00 the $HNO_3$ mixing ratio
decreases slowly throughout the flight, which is the result of $HNO_3$ generation in the $^{210}Po$ source leading to an initially large
background signal. The formation of $HNO_3$ in the $^{210}Po$ source has been documented previously (Ji et al., 2020); its level can
be reduced by permanently flushing $N_2$ through the source while keeping the mass-spectrometer under operational vacuum.
This was not possible during the CAFE missions on HALO as continuous operation of the instrument (i.e. overnight between
flights) was not possible. A roughly exponential decay of the $HNO_3$ background signal was observed in all of the flights in
which $m/z$ 62 was monitored, which presumably reflects depletion of the initially large $HNO_3$ reservoir which was built up
when the I-CIMS was switched off.

A rough correction of the dataset was thus undertaken by subtracting an exponentially decaying background from the total
$HNO_3$ signal. The resulting $HNO_3$ mixing ratios are depicted as the blue line in Fig. 9b and plotted against the $O_3$ mixing
ratio in Fig. 9c. Considering only the high altitude data for which $O_3$ mixing ratios were > 100 ppbv (stratospheric influence,
black data points) we derive a slope of $HNO_3$ / $O_3$ = $(3 \pm 0.5) \times 10^{-3}$ (the uncertainty is 2 σ, statistical only) which is
consistent with previously reported values obtained in airborne measurements of $HNO_3$ and $O_3$ in the lower stratosphere (see
Popp et al. (2009) and references therein). We stress that deriving accurate mixing ratios of $HNO_3$ is not possible with this
data set and the values obtained are strongly dependent on the background correction. Here, we merely wish to indicate that,
while most of the variability in our $m/z$ 62 signal is related to the central role of ozone in the detection scheme (i.e. formation
of $IO_X$), some covariance between $HNO_3$ and $O_3$ remains after correction of the raw data and the slope is roughly in line with
that expected. We also do not propose that the correlation of $m/z$ 62 with $O_3$ proves that the signal can be attributed entirely
to $HNO_3$. This aspect will be covered in section 4.2.

Examining Fig. 9b reveals sharp increases in the (background corrected) $HNO_3$ mixing ratio when sampling at lower
altitudes, noticeably at 11:30- 12:00 (3.9 km altitude) and at 15:45-16:10 (4.7 km altitude) and at the end of the flight during
descent to Oberpfaffenhofen in Bavaria, Germany. In all cases, these periods of enhanced $HNO_3$ coincided with higher levels
of particles. Back trajectories (HYSPLIT) indicated that, in the 10 days prior to interception by HALO, the air mass sampled
at 11:30 had passed over the West African continent (Mauritania, Mali and Niger), whereas the air masses sampled after





16:00 were of European origin. The large, coincidental increase in the $HNO_3$ mixing ratio and particle mass was a recurrent feature of the CAFE-Africa flights. It is conceivable that the $HNO_3$ measured by the I-CIMS was a mixture of gas-phase $HNO_3$ and $HNO_3$ associated with particles that desorb $HNO_3$ when passing through the thermal dissociation inlet at 180 °C. This temperature would be sufficient to thermally convert ammonium nitrate to $HNO_3$ (and $NH_3$) as well as to result in the

desorption of $HNO_3$ that was physi-sorbed e.g. on chemically aged black-carbon or mineral-dust particles. As we do not know the efficiency with which particles of various diameters enter the TD-inlet of the CIMS, we cannot estimate the relative contribution of gas-phase and particulate nitrate to the signal at $m/z$ 62 but indicate that a similar phenomenon may occur in ground-based measurements using TD-inlets and may represent an additional source of bias during ambient measurements of $NO_3$ and/or $N_2O_5$ at $m/z$ 62.

**4.2 CAFE-Europa**

During the CAFE-Europe HALO flights the I-CIMS monitored $m/z$ 190, the $I^-(HNO_3)$ adduct, as well as $NO_3^-$ at $m/z$ 62. The detection of $HNO_3$ at $m/z$ 190 is not sensitive to the $O_3$ mixing ratio but does vary with the water vapour concentration in the IMR. The response of the $HNO_3$ signal at $m/z$ 190 to changes in the $HNO_3$ concentration and in the $m/z$ 145 / $m/z$ 127 ratio (i.e. the relative humidity in the IMR, see Fig. S2) is illustrated in Fig. S5.

Figure 10 displays a set of data obtained during a flight on the 30th May 2020 on which the HALO aircraft flew a path from Southern Germany to the Atlantic (west of Ireland) and back at various altitudes (for flight track see Fig S6). Figure 10a plots the raw signals measured by the I-CIMS at $m/z$ 62 and $m/z$ 190 as well as the $O_3$ mixing ratio. Similar to the CAFE-Africa data-set, the signal at $m/z$ 62 covaries strongly with the $O_3$ mixing ratios, which were between ~40 and ~700 ppbv. The signal at $m/z$ 190 does not show any correlation with $O_3$ and the raw signals at $m/z$ 62 and $m/z$ 190 bear little

resemblance to each other.

Using the calibrations parameters described in section 3 and (for $m/z$ 190) in Fig. S4, the signals at $m/z$ 62 and $m/z$ 190 were converted to $HNO_3$ mixing ratios, depicted in Fig. 10b. Despite the greatly divergent raw-signals, the $HNO_3$ mixing ratios obtained using the different mass-to-charge ratios are in good agreement, both displaying a gradual decrease after take-off at ~08:00 UTC. The high initial level of $HNO_3$ is largely the result of $HNO_3$ being formed in the $^{210}Po$ source during overnight

instrument shut-down (see section 4.1). The $HNO_3$ mixing ratios observed at $m/z$ 62 and $m/z$ 190 both increase when the aircraft sampled stratospheric air (11:00 to 13:00 and 15:10-15:30 UTC). In Fig. 10c $HNO_3$ mixing ratios derived at $m/z$ 62 and $m/z$ 190 are plotted in a correlation diagram. The slope (0.98 ± 0.05) and intercept (-0.01 ± 0.28) indicate good agreement even when the raw signals are highly divergent at high levels of $O_3$. The provides strong evidence that, in many if not most air masses, $m/z$ 62 provides a measure of $HNO_3$ rather than $NO_3$ and $N_2O_5$.

**5 Conclusions**

A series of laboratory experiments investigating the origin of signal at *m/z* 62 when using an I-CIMS has revealed unexpected sensitivity to $HNO_3$ at this mass-to-charge ratio in the presence of $O_3$ or peracetic acid (PAA) or PAN. The ozone effect is related to the formation of $IO_X^-$ which react rapidly with $HNO_3$ to form $NO_3^-$ thus bypassing the thermodynamic barrier to formation of $NO_3^-$ by direct reacton of $HNO_3$ with $I^-$. The presence of $O_3$ at a mixing ratio of 500

ppbv results in a 250-fold increase in sensitivity to $HNO_3$ at *m/z* 62. The sensitivity to $HNO_3$ at this mass-to-charge ratio was also found to be highly dependent on the concentration of $H_2O$ in the ion-molecule reactor as this aids formation of $IO_X^-$. The sensitivity to $HNO_3$ at *m/z* 62 in the presence of PAA is a result of the presence of acetate anions ($CH_3CO_2^-$) as demonstrated previously (Veres et al., 2008). We suggest that measurements of PAN using I-CIMS may be biased to low values if large mixing ratuios of $HNO_3$ (or organic acids) are present. Our laboratory experiments indicate that measurements of

atmospheric $NO_3$ and $N_2O_5$ at *m/z* 62 can be heavily biased by the presence of $HNO_3$, and may explain reports of unexpectedly high daytime mixing ratios of $N_2O_5$. The relaive sensitivity at *m/z* 62 to $HNO_3$ and $N_2O_5$ / $NO_3$ will vary from one I-CIMS instrumet to the next and must thus be analysed case-by-case.

We have examined signals at *m/z* 62 during two periods of operation of the I-CIMS on the HALO-aircraft, one over the Atlantic west of the African coast and one over Europe. During the flights over Europe $HNO_3$ mixing ratios derived from *m/z*

62 ($NO_3^-$) and at *m/z* 190 ($I^-(HNO_3)$) were in very good agreement. The data obtained over the Atlantic indicated that measurements at *m/z* 62 using a thermal dissociation inlet can be strongly influenced by particulate nitrate that can thermally dissociate (or desorb) to gas-phase $HNO_3$.

**Data availability**

Data measured during the flight campaign CAFE campaigns are available to all scientists agreeing to the CAFE data

protocol. The laboratory data underlying the Figures is availlable upon request to the authors.

**Author contributions**

RD conducted the laboratory experiments, carried out the airborne measurements with assistance from PE and JC and analysed the laboratory with assistance from JC. The manuscript was written by JC and RD with contributions from all other authors. JL designed and helped plan the airborne operations.

**Competing interests**

The authors declare that they have no conflict of interest.



## Acknowledgments

We acknowledge the collaboration with the DLR (German Aerospace Centre) during the HALO campaigns CAFE- Africa and CAFE-EU. We thank Florian Obersteiner and Andreas Zahn (KIT-Karlsrühe) for use of the $O_3$-data during CAFE-Africa and CAFE-EU.

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





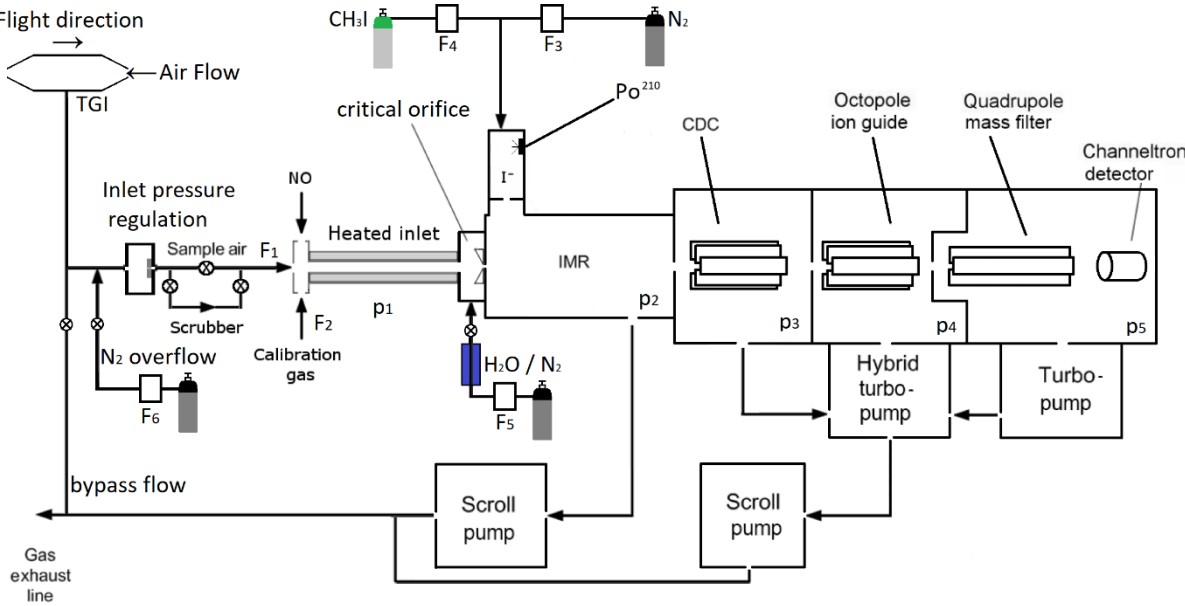


**Figure** 1 Schematic diagram illustrating the central components of the I-CIMS used in this work. IMR = ion-molecule reactor, CDC = collisional dissociation chamber, $F_1$ = 1250 cm$^3$ (STP) min$^{-1}$, $F_2$ = 50 cm$^3$ (STP) min$^{-1}$, $F_3$ = 750 cm$^3$ (STP) min$^{-1}$, $F_4$ = 4 cm$^3$ (STP) min$^{-1}$, $F_5$ = 50 cm$^3$ (STP) min$^{-1}$. p1 = 100 mbar, p2 = 24 mbar, p3 = 0.6 mbar, p4 = 6 × 10$^{-3}$ mbar, p5 = 9 × 10$^{-5}$ mbar. The heated inlet is made of PFA-tubing. TGI = Trace-gas-inlet.




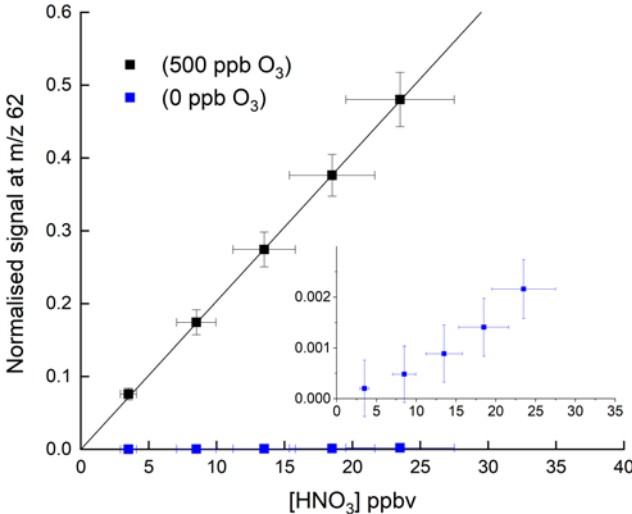


**Figure 2.** HNO$_3$ detection at *m/z* 62 in the absence and presence (500 ppbv) of O$_3$. The solid lines are non-weighted, linear regressions to the data and are $2.035 \times 10^{-2}$ and $8.033 \times 10^{-5}$ ppbv$^{-1}$ HNO$_3$ when 500 ppbv O$_3$ or zero O$_3$ were present, respectively. The inset (same x- and y-axes as in the full figure) is an expanded view of the signal obtained in the absence of

O$_3$. The error bars represent 15% systematic uncertainty in the HNO$_3$ concentration and 2$\sigma$ statistical uncertainty in the signal at *m/z* 62.






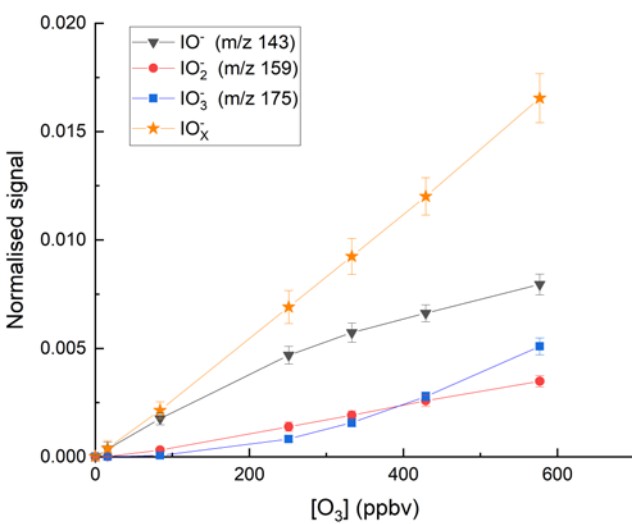

**Figure 3**: Variation of the I-CIMS signals at *m/z* 143 (IO⁻),  159 (IO₂⁻) and 175 (IO₃⁻)with the mixing ratios of $O_3$. The $O_3$ mixing ratios are those measured in air before the gas-flow entered the inlet. The water vapour was held constant using our

standard setting ($[H_2O]_{IMR} = 2.9 \times 10^{14}$ molecule cm⁻³).





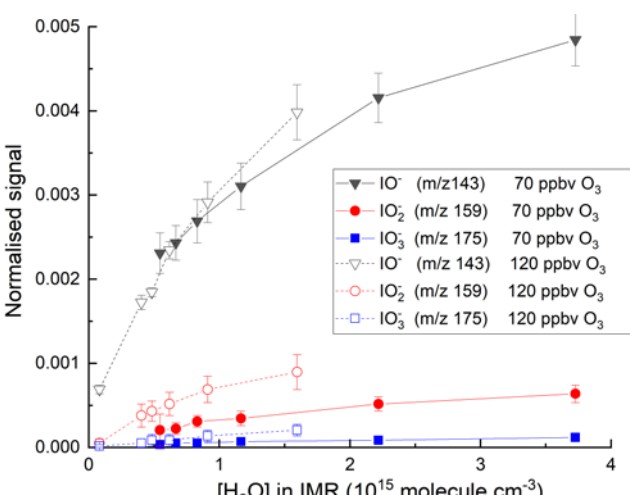

**Figure 4:** Variation in the total ion signal (counts) due to $IO^-$, $IO_2^-$ and $IO_3^-$ with the concentration of water vapour in the

IMR. The results are from two sets of experiments where the $O_3$ mixing ratio was either 70 or 120 ppbv.





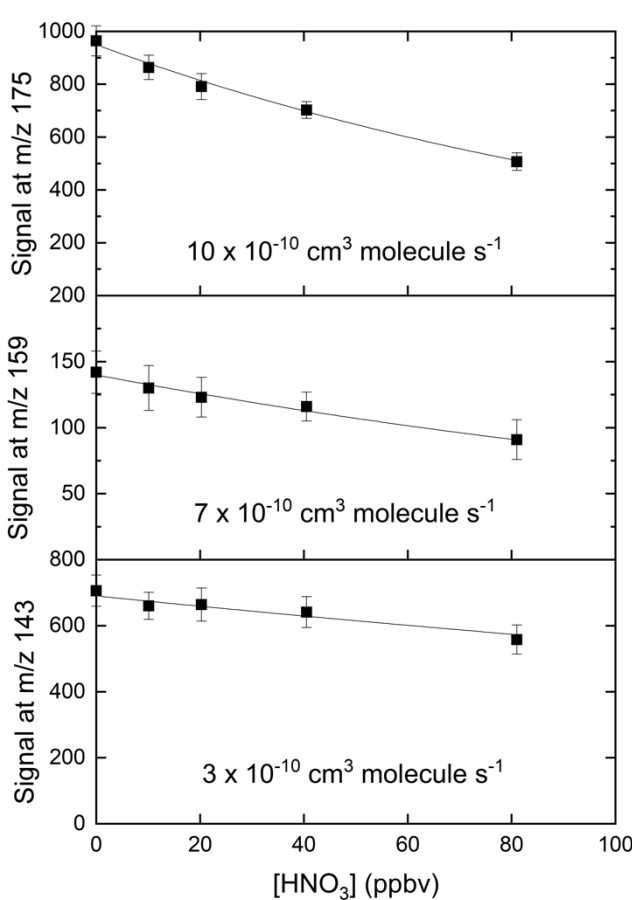

**Figure 5:** Relative changes in signals at *m/z* 143 (IO⁻), *m/z* 159 (IO₂⁻) and *m/z* 175 (IO₃⁻) when adding up to 80 ppbv of HNO₃. The rate coefficients were calculated using a reaction time of 25 ms and should thus only be regarded as approximate.





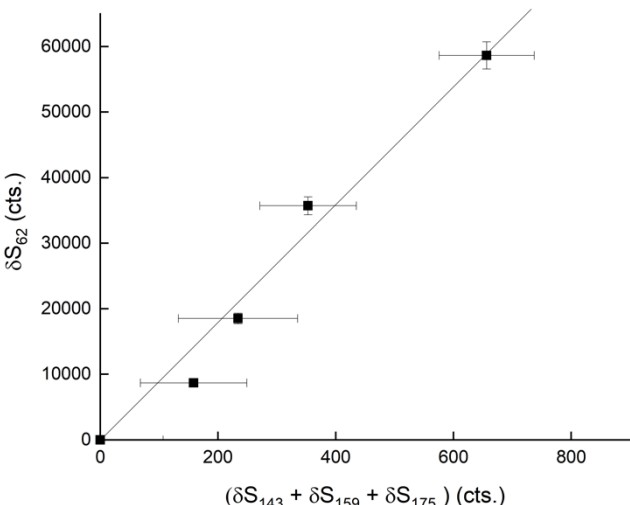

**Figure 6:** Relative changes in the sum of signals at *m/z* 143 (IO⁻), *m/z* 159 (IO$_2$⁻) and *m/z* 175 (IO$_3$⁻) when adding up to 80 ppbv of HNO$_3$. δ refers to the change in signal upon adding HNO$_3$ and thus takes background signals at each mass-to-charge ratio into account.






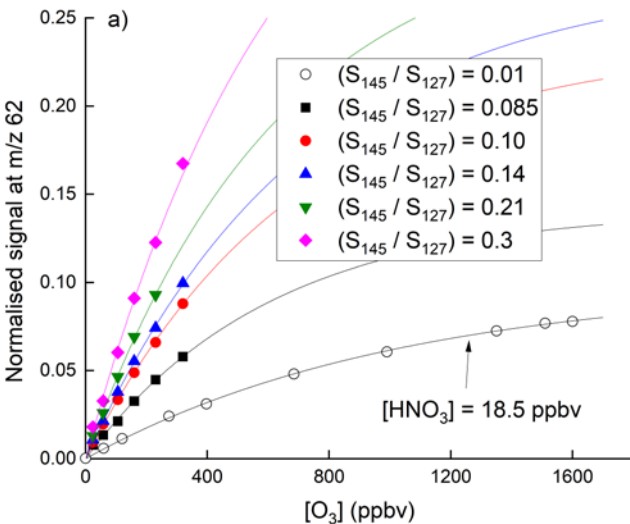

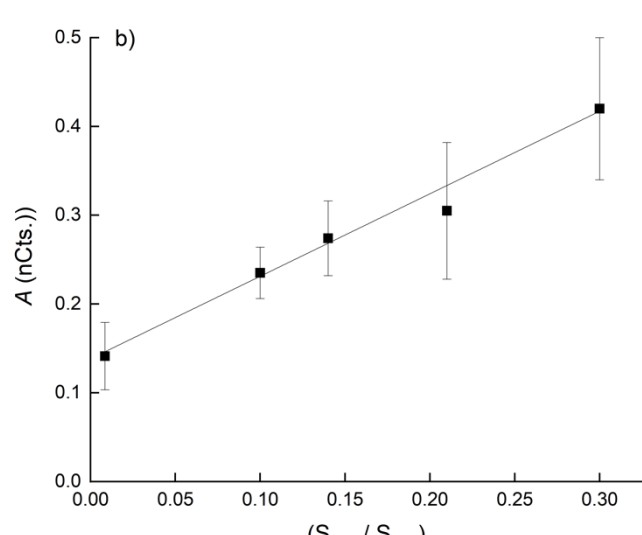

**Figure 7**. a) Dependence of the signal at *m/z* 62 on the $O_3$ concentration for 6 different concentrations of $H_2O$ in the IMR. In the upper 5 curves (solid symbols) the $HNO_3$ mixing ratio was 38.5 ppbv. In the lowermost curve, the $HNO_3$ mixing ratio

was 18.5 ppbv as indicated. The fits lines are of the form: y = $A$*exp(1-exp(-$B$*[$O_3$])). b) Plot of parameter $A$ versus the relative signal at *m/z* 145 and *m/z* 127.





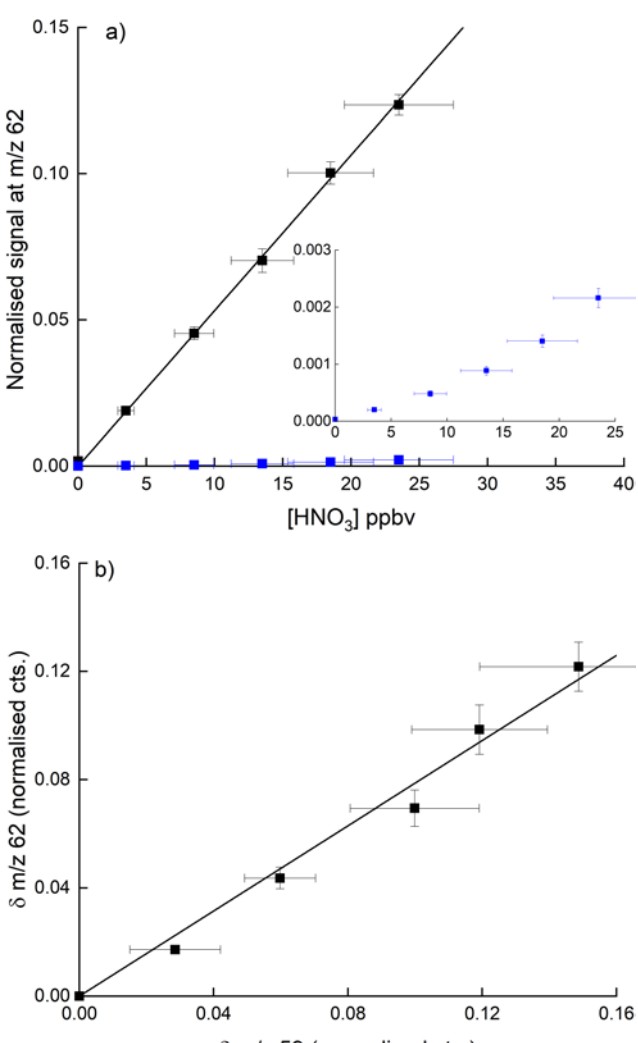


**Figure 8**. a) Detection of HNO$_3$ at *m/z* 62 in the presence of 3.25 ppbv PAA (and thus the acetate anion, CH$_3$CO$_2$). The blue data points (expanded view in the inset) were obtained in the absence of PAA, whereby detection of HNO$_3$ at *m/z* 62 is inefficient. The error bars are 1σ statistical uncertainty in the signal at *m/z* 62 and 17 % total uncertainty in the HNO$_3$ mixing ratio. (b) Change in normalised signals at *m/z* 62 and *m/z* 59 upon adding HNO$_3$ for the same dataset (i.e. background

corrected signals). The error bars are 1σ statistical uncertainty.





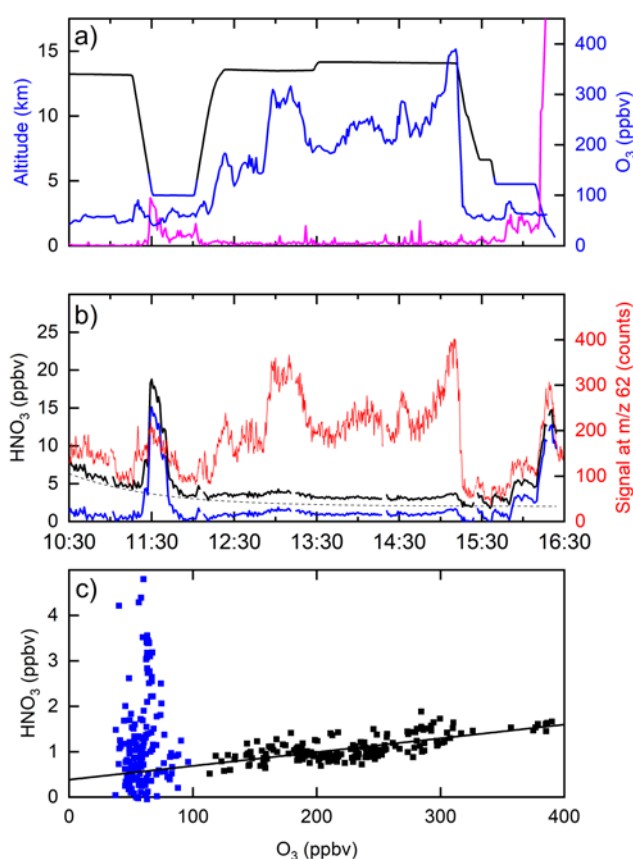

**Figure 9.** a) Altitude (black) and $O_3$ mixing ratios (blue) from a HALO-flight during the CAFE-Africa campaign. The
purple line (arbitrary units) is proportional to the black-carbon particle number density. b) The signal at *m/z* 62 (red line)
clearly co-varies with $O_3$. Following conversion to a mixing ratio (black line) and subtraction of a $HNO_3$ background signal
(dotted line) originating in the $^{210}Po$-source, the solid blue line for $HNO_3$ is obtained. c) $HNO_3$ mixing ratios plotted versus
$O_3$ mixing ratios. The straight black line has a slope of $(3 \pm 0.5) \times 10^{-3}$ and does not take into account the blue datapoints ($O_3$
mixing ratio < 100 ppbv).




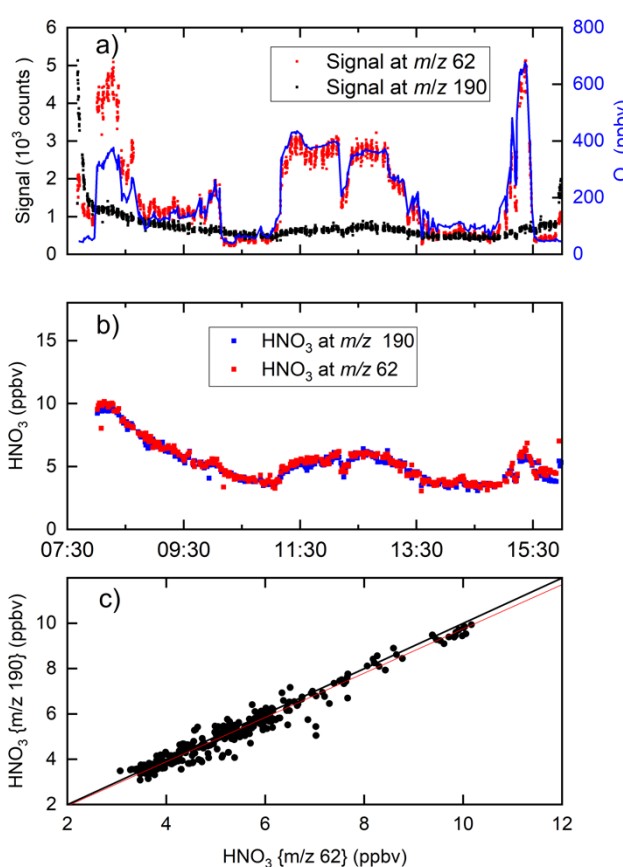

**Figure 10**. I-CIMS HNO$_3$ measurements and auxiliary data from a HALO-flight during the CAFE-EU campaign. a) Signals at $m/z$ 62 and $m/z$ 190 as well as O$_3$ mixing ratios. b) HNO$_3$ mixing ratios derived from the signals at $m/z$ 62 and $m/z$ 190 taking the dependence of sensitivity on ozone ($m/z$ 62) and relative humidity ($m/z$ 62 and $m/z$ 190) into account. c) Correlation of the HNO$_3$ mixing ratios derived from the two masses. The red line is a bivariate fit (slope 0.98 ± 0.05, intercept -0.01 ± 0.28). A large fraction of the HNO$_3$ measured stems from the polonium source, especially at the beginning of the flight.