# Peer review of "Iodide-CIMS and m/z 62: The detection of HNO3 as NO3- in the presence of PAN, peroxyacetic acid and ozone"

_Atmospheric Measurement Techniques, 2021_

## Author Comment (AC2)

**Iodide-CIMS and *m/z* 62: The detection of HNO$_3$ as NO$_3^-$ in the presence of PAN, peracetic acid and O$_3$**

Raphael Dörich,[1] Philipp Eger,[1] Jos Lelieveld[1] and John N. Crowley.[1]

[1]Atmospheric Chemistry Department, Max Planck Institute for Chemistry, 55128, Mainz, Germany

*Correspondence to*: John N. Crowley (john.crowley@mpic.de)

**Supplementary Information**

[Figure]

**Figure S1**. Signal at *m/z* 62 and O₃ mixing ratios during airborne operation of the I-CIMS during the HALO campaign "CAFE-Africa".

[Figure]

**Figure S2**. Dependence of the ion-signal ratio *m/z* 145 / *m/z* 127 (corresponding to $I^-(H_2O)$ and $I^-$, respectively) on the degree of humidification of zero-air entering the inlet. The relative humidity (RH) was monitored with a hygrometer (Testo 625) at a pressure of 1 bar and a temperature of 298 K. The regression line is: RH = -24.616 *ln*{0.969 – (1.923 × $S_{145}$ / $S_{127}$)}.

Using the expression above RH can be converted to a mixing ratio (MR), pressure (P, in mbar) or concentration (C, in molecule $cm^{-3}$) of $H_2O$ in the IMR via:

$$MR = \frac{\left(\frac{RH}{100} \times 31.7\right)}{1000} \times \frac{1200}{2000}$$

where 31.7 is saturation vapour pressure (mbar) of $H_2O$ at 298 K, 1000 is ambient pressure (mbar), 1200 and 2000 are the humidified and total flows (in sccm), respectively into the IMR.

$P(H_2O) = MR \times 24$

Where 24 is the pressure (in mbar) in the IMR

$C(H_2O) = P(H_2O) \times 2.43 \times 10^{16}$

Where $2.43 \times 10^{16}$ is a conversion factor for mbar into molecule $cm^{-3}$ at 298 K

[Figure]

**Figure S3.** Dependence of the signal at *m/z* 62 from 38.5 ppb of $HNO_3$ on the amount of $H_2O$ in the IMR (as indicated by the ratio of signals of $I^-$ (*m/z* 127) and its water cluster (*m/z* 145). The error bars are $1\sigma$ statistical uncertainty in the signal at *m/z* 62, *m/z* 127 and *m/z* 145

[Figure]

**Figure S4**. Flight track (colour coded with altitude) during the return leg (Cap Verde Islands to Oberpfaffenhofen) of the HALO-campaign "CAFE-Africa". The numbers on the track indicate the time (UTC).

[Figure]

**Figure S5**. Upper panel: Dependence of the signal at *m/z* 190 due to the I⁻(HNO₃) cluster on the HNO₃ mixing ratio . Lower panel: Dependence of the signal at *m/z* 190 on the relative humidty ($S_{145}$ / $S_{127}$).

[Figure]

**Figure S6**. Flight track (colour coded with altitude) during a flight (30.05.2020) over Europe of the HALO-campaign "CAFE-EU". The numbers on the track indicate the time (UTC).

---

## Author Comment (AC3)

**Referee 1**

We thank the referee for a positive assessment of our manuscript. The comments (in black), our response (in blue) and changes to the manuscript (in red) are listed below.

R. Dörich and co-workers have presented convincing laboratory evidence that iodide-CIMS instruments efficiently detect HNO3 as NO3- but ONLY in the presence of ozone (which converts I- into IOx- ions, which in turn react with HNO3). This finding has substantial implications for field studies using such instruments, and suggests that some previous measurements have been incorrectly interpreted. This is one more example of how the powerful tool of chemical ionisation should be used very carefully, with due consideration of possible side reactions, including indirect pathways such as that discovered here. The study is definitely worth publishing in AMT. I have only very minor corrections and questions as described below.

-On line 135, reaction (R6) should presumably be reaction (R7), i.e. I- + HNO3 not I- + H2O.
Correction made

-On line 153, the product should presumably be IO3- not IO2-.
Correction made

-Line 159: Maybe mention already here the the O2 concentration in the quoted studies was MUCH less than that in the atmosphere (or in these measurements) - I had a hard time reconciling the dominance of IO3- with the stated rate coefficients, since I kept assuming 0.2 atm $O_2$. Also, is it the lack of an IO3- + O2 reaction that drives the equilibrium toward IO3- ?
We now mention that the $O_2$ concentrations were ~4x that of $O_3$. The question of what changes the equilibrium concentrations of IOx in our system is addressed later in the manuscript.
With the $O_3$ (~1-5 × $10^{10}$ molecule cm$^{-3}$), $O_2$ concentrations (~ 4 × that of $O_3$) and reaction times used in these studies...

Could the authors use e.g. gas-phase acidity / proton affinity data to estimate thermodynamic parameters (at least endo/exothermicity) for reactions R13-R15 (and also R19-R21)?
We have add some information regarding thermodyanic properties for the "new" reaction of $IO_x^-$ with $HNO_3$:
Using heats of formation (in kJ mol$^{-1}$ at 298 K) of $\Delta H_f(IO_3^-)$ = -211 (Eger et al., 2019), $\Delta H_f(HNO_3)$ = -134 (Goos et al., 2005), $\Delta H_f(HOIO_2)$ = -95 (Khanniche et al., 2016) and $\Delta H_f(NO_3^-)$ = -312 (Goos et al., 2005) we calculate that reaction R15 is exothermic by ~62 kJ mol$^{-1}$.
The reactions listed in R19-R21 are well known and documented as CIMS detection schemes; in this case the addition of thermodynamic properties is not necessary.

Could the authors speculate about the reasons for the differences in rate coefficients for reactions R13…R15? The ion size seems to play a role, but is that enough to explain a difference of a factor of 3 between IO- and IO3-?
We state that "The results indicate qualitatively that $IO_3^-$ is the most reactive of the $IO_X^-$ anions towards $HNO_3$, but that all three contribute to $HNO_3$ detection." As the referee states, ion-size may play a role but there will be other factors. Given the assumptions made in deriving these "approximate" rate coefficients, and our lack of theoretical tools to examine ion-molecule reactions in detail, we feel that discussion of the apparent differences in rate coefficients is not appropriate.

-Line 221: "shut of" should be "shut off"
Correction made

---

## Author Comment (AC4)

**Referee 2**
We thank the referee for a positive assessment of our manuscript. The comments (in black), our response (in blue) and changes to the manuscript (in red) are listed below.

This is a very nice manuscript that explores a large dynamic signal that occurs at mass to charge ratio 62 (NO3-) in the iodide CIMS instruments. This signal has long puzzled users of the iodide ion for detection of trace gases, and this manuscript adds extremely valuable information to that ongoing debate. The manuscript is clearly written and organized and presents a well thought out convincing argument assigning signal at 62 to HNO3 detection via the IOx- anion. This will be a valuable addition to the existing literature on iodide CIMS and I fully support its publication pending minor changes.

It has been my own experience, having run a similar set of experiments, that addition of significant O3 to the iodide CIMS instrument while holding HNO3 addition constant will result in observable depletion of IHNO3- observed at m190, even when normalizing to iodide. The authors convey nicely that the impact of ozone on signal detection will be highly variable depending on the instrumental conditions used, and that alone could be the difference here. However, considering the authors present HNO3 data from the field using observations at m/z190, the data should exist from the laboratory experiments to validate that this does not occur in their system. It would be beneficial to this manuscript if the authors could include a figure similar to 7a but looking at the signal at m/z 190. This information is important to validate some of the assumption throughout the manuscript such as on page 8, line 232 where it is indicated that "negligible depletion of HNO3" occurs, a fully testable assumption considering the ability to detect HNO3 at m/z 190. It would also be useful in understanding the comparisons of the m62 and m190 data at the end of the manuscript from the ambient observations.

In summary, the referee asks whether the $HNO_3$ mixing ratio in the IMR (and thus its detection at $m/z$ 190) is influenced by the addition of $O_3$. We have conducted laboratory experiments in which a constant flow of $HNO_3$ was monitored at $m/z$ 62 and $m/z$ 190 while various amounts of $O_3$ (up to 800 ppbv) were added. The results are now displayed as a new Figure (8) and described in a new section (3.1);

**3.2 Detection of HNO₃ at *m/z* 190**

We now compare the detection of $HNO_3$ at $m/z$ 62 to its detection at $m/z$ 190, the $I^-(HNO_3)$ adduct, with various amounts of $O_3$ present. In Fig. 8 we present the results of an experiment in which a constant flow of $HNO_3$ (12.5 ppbv) was introduced into the inlet and the ozone mixing ratio was varied from zero to 900 ppbv. We observe a great increase in the signal at $m/z$ 62 as expected (from 9 counts to > 6000 counts). At zero ozone, the signal at $m/z$ 190 is about 1000 counts and is largely background free, making this the preferred mass to monitor $HNO_3$ in the absence of $O_3$. The cross-over point (when the signals at $m/z$ 62 and $m/z$ 190 are equal) is at an ozone mixing ratio of 100 ppbv. At an ozone mixing ratio of 800 ppbv, the signal at $m/z$ 62 is a factor 8.5 larger than that at $m/z$ 190.

Apparent from this figure is the depletion of the signal at $m/z$ 190 as the $O_3$ mixing ratio increases to values of 800 ppbv, as observed e.g. in the lower stratosphere. The solid lines are least-squares fits to the datasets that describe the exponential growth of the $m/z$ 62 signal as $O_3$ is increased and the exponential decay of $m/z$ 190 over the same range. The reduction in signal at $m/z$ 190 is characterised by an exponential term $\exp(-0.00046 \times [O_3])$ which means that at 800 ppbv $O_3$ a ~30% reduction in sensitivity is observed. The loss of sensitivity at $m/z$ 190 is driven by the loss of $I^-$ in the IMR as $O_3$ is added. At the same time, the depletion of the signal at $m/z$ 127 is weaker, which reflects the fact that only a small fraction of $IO_X^-$ formed in

the IMR are detected, the rest presumably being fragmented in the CDC before being detected as $I^-$.

The fact that some $I^-$ is converted to $IO_X$ in the IMR at high $O_3$ levels but is not reflected in the $I^-$ signal at $m/z$ 127 has repercussions for normalisation of product ion signals to the primary ion signal whereby the assumption is made that the measured signal at $m/z$ 127 stems only from $I^-$. In our instrument, the loss of $I^-$ in the IMR is significant at high levels of $O_3$ (e.g. 30 % at 800 ppbv $O_3$). For this reason, we normalise the signals using values of the signal at $m/z$ 127 obtained by interpolating between measurements obtained when scrubbing the air. The normalisation problem may occur in other I-CIMS instruments to varying extents, and the degree of bias will depend on the conversion of $I^-$ to $IO_X^-$ and the extent to which $IO_X^-$ is detected as $I^-$. The potential bias can be circumvented by the continuous addition of a calibration gas that is detected via reaction with $I^-$.

We have modified text in other positons. On page 5 we now write:
Throughout the paper, when presenting raw data, we generally normalise the I-CIMS signal by dividing by the primary ion signal at $m/z$ 127. This is standard practise and corrects for drifts in the $CH_3I$ flow which may occur over several hours after the instrument was switched from standby mode to operational mode. It also accounts for longer term drifts caused by the weakening activity of the $^{210}Po$ source over the duration of a measurement campaign (months) or since the last calibration and for loss of detector sensitivity over similar time periods. We show later, that when adding large concentrations of reactants that significantly deplete the primary ion signal at $m/z$ 127 this procedure may lead to bias is some measurements. For this reason, when detecting $HNO_3$ at $m/z$ 190 (see later), we normalise to an interpolated signal at $m/z$ 127 that was measured when the air was scrubbed.
In the abstract we now write:
At high ozone mixing ratios, we show that the concentration of $I^-$ ions in our IMR is significantly depleted. This is not reflected by changes in the measured $I^-$ signal at $m/z$ 127 as the $IO_X^-$ formed do not survive passage through the instrument, but are likely detected after fragmentaton to $I^-$. This may result in a bias in measurements of trace gases using I-CIMS in stratospheric air masses unless a calibration gas is continuously added or the impact of $O_3$ on sensitivity is characterised.
Figure 10 (now 11) has been modified to take the $O_3$ dependence of the sensitivity to $HNO_3$ at $m/z$ 190 into account.

I am having a difficult time understanding how the authors are able to retrieve an ambient HNO3 signal from the NO3- signal. The signal that occurs at that mass is dependent on a changing HNO3 ambient concentration as well as a changing O3 concentrations. It would seem rather difficult to retrieve the real concentration unless you make an assumption that at any given point HNO3 is not changing and only O3 is changing, or the other way around. The only way this may work is if the authors used a measurement of true O3 collected during the ambient flights, however it is unclear from the description that this was the case, perhaps those details are just missing. If an ozone measurement was used what instrument provided that data. I would like to see more details of how the concentration profiles in Figures 9 and 10 are determined, and what assumptions were made. If the author agree that one can not retrieve true HNO3 unless a true ozone measurement is used it would benefit the readers to explicitly state this. It is possible that a reader could interpret the data presented here to mean that by using the Iodide CIMS measure of IOx- and NO3- one can quantitatively measure HNO3.
It is, of course, correct that $HNO_3$ can only be derived from the signal at $m/z$ 62 if $O_3$ (and $H_2O$) are simultaneously measured. However, these parameters are essential to analysis of

atmsopheric composition and are measutred routinely in the HALO aircraft. We now mention this is the manuscript.

During both HALO campaigns, $O_3$ and $H_2O$ (required for analysis of the signal at $m/z$ 62) were routinely measured.

I am in general surprised by the lack of instrument response to ambient features in the data in figures 9, 10 and S1. HNO3 should have a significant inlet response time relative to O3 in the CIMS regardless of the inlet conditions. In figure S1 in particular there appears to not be any lag in the observations. This quality in addition to the exact correlation to O3 makes me believe that you are really just detecting HNO3 that is made inside of the instrument. Otherwise, I would expect some deviation between the ambient O3 and HNO3 from 12:00 to 15:00 in figure S1. Can the authors comment on the time response of their measurements? What does a laboratory time response on m190 look like relative to m62?

An examination of Figure 9 (blue $O_3$ curve in panel a and red $m/z$ 62 curve in panel b) shows that featues (e.g. in $O_3$) are matched by those in $m/z$ 62. The "time lag" which the referee refers to presumably relates to the sticky nature fo $HNO_3$ on inlet material and might be expected to occur (to an extent depending on the inlet material and temperature) if the $HNO_3$ and $O_3$ were both ambient. We agree with the statement that the dominant contribution to the signal at $m/z$ 62 is from the polonium source, and we clearly show this in Figure 9. We do not try to disguise this fact and state that (because of this) the $m/z$ 62 data is not accurate and should not be overinterpreted. Comparison with the time-response of a laboratory source of $HNO_3$ in not helpful in this regard as we cannot conduct lab experiments under flight conditions using the HALO-trace-gas-inlet.

Has the data from Figure 10b been corrected for scrubber zeros? Is this actually signal over zero. The way the data correction is explained for 9b where an approximate exponential of the Po HNO3 source is subtracted out would suggest that there are not real zeros removed from the ambient data, which would include source HNO3. If real zeros have not been used how do you know any of the features shown in figure 9b or 10b are real ambient signals. If real zeros were used why didn't they account for the Po source background?

The data presented in figure 10b has not been corrected for background $HNO_3$ from the Polonium source. This is not possible as scrubbing also removes $O_3$ and thus the signal from $HNO_3$ generated in the polnium source would also be (falsely) zeroed. We now clarify this and write:

We note that the background signal at $m/z$ 62 that originates from the polonium source cannot be obtained by scrubbing the air of $HNO_3$ as this also removes $O_3$ and thus also sensitivity to $HNO_3$ at this mass.

This ion chemistry is ultimately a three-body reaction as indicated in page 7, line 216. Therefore, the result should be quite dependent on IMR pressure. The iodide CIMS literature is filled with instruments running across a very large range of pressures, sometimes exceeding 100 mbar, do the authors have any data or comments on the impact of pressure on this ionization mechanism.

On page 7, line 16 we listed a NET reaction sequence in which three molecules are involved in sequential, bimolecular reactions. This is not to be confused wuth a termolecular (3-bodied reaction) in which pressure (i.e. collision rates) plays a role in quenching adducts that are initially formed with some degree of internal excitation. There is no reason to assume that the chemistry we outline is pressure dependent. In any case, as our objective was to understand the signals obtained during the HALO campaigns, we only conducted experiments at our "standard" airborne IMR pressure.

The authors comment on the potential issues with past data sets using iodide CIMS to measure PAN, due to the secondary acetate ion chemistry that is occurring. It is my experience that many if not all of the airborne systems employ a constant standard addition of 13C PAN to the instrument. I believe that this addition would account for any secondary ion loss due to IOx-chemistry. I think the authors should add a comment in their discussion on this point on pages 9/10. While there are likely many dataset that have not used such an internal standard to track sensitivity changes it is not appropriate to question the data set that have used that method. A related question is do the authors believe such a system would indeed correct for these ion chemistry issues? That could be a good couple sentence discussion to add here.

The referee is correct in stating that a continuous, internal 13C standard will avoid this problem (assuming that labelled acetate anons reacts with the same rate coefficient as the unlabelled one). We have amended the text appropriately:

One way to avoid this problem is the continuous addition of isotopically labelled PAN to the inlet  (see e.g. Roiger et al. (2011)) as the secondary, reactive losses of  $^{12}C$ and $^{13}C$ $CH_3CO_2^-$ are expected to be identical.

And also the abstract:

The loss of $CH_3CO_2^-$ via conversion to $NO_3^-$  in the presence of $HNO_3$ may represent a significant bias in I-CIMS measurements of PAN and $CH_3C(O)OOH$ in which continuous calibration (e.g. via addition of isotopically labelled PAN) is not carried out.

On page 12, line 369 the authors state that 190 does not show any correlation with O3. However, in the same figure, #10, it is clear that 190 correlates with m62. It was previously argued in describing figure 9, using panel C that it is expected that HNO3 should correlate with O3 even after correction for this unique ion chemistry. Additionally, there are clearly features in 10b that correlate with the ozone signal shown in 10a. Please elaborate on the disagreement between these two statements.

The statement was ambiguous as correlation can be caused both by the expected covariance of stratospheric $HNO_3$ and $O_3$ but (for $m/z$ 62) also stems from the strong dependence of the sensitivity to $HNO_3$ on the $O_3$ mixing ratio at this mass. We now write:

Similar to the CAFE-Africa data-set, the signal at $m/z$ 62 covaries strongly with the $O_3$ mixing ratios, which were between ~40 and ~700 ppbv whereas the raw signals at $m/z$ 62 and $m/z$ 190 (both due to $HNO_3$) bear little resemblance to each other.

On page 6 there is a discussion on the residence time in the flow tube and comparison to similar work. I encourage the authors to reconsider their calculation of the residence time in their flow tube as presented. With a critical orifice on a cylinder there will be a jetting effect of the air through the region such that your reaction timescale will not be equivalent to the laminar sweeping of the volume. Rather it will be dependent on the velocity and flow dynamics of that jetting effect. This likely results in a significantly shorter reaction time than the volumetric calculation for most of the analyte molecules being sampled. It is not necessarily a discussion that is needed here but should be considered when trying to understand the difference being discussed, and leveraged as a potential answer for some of the disagreement observed.

Agreed. We do not know the flow-dynamic in the IMR and now emphasise that the reaction times are rough approximations. We write:

Note that the IMR reaction times we derive are only approximate as we do not take into account the mixing and flow dynamics in the IMR, which are likely to be complex (and possibly shorter than 25 ms) owing to sampling via a critical orifice. While we can not rule out that our observation of IO$^-$ (and not IO$_3^-$) being the dominant ion-signal is partially caused by differences in reaction times, slight differences in $O_3$ concentrations and differences in temperature (our IMR is at ~15°C above ambient temperature owing to the heated inlet) we note that the higher pressure of $O_2$ (factor ~1-10 × 10$^4$) in our IMR is likely to have a large effect.

Figure 1. I am interested in the flow dynamics in the figure with the initial sampling line. It appears that at some point your inlet could be connected to a pump exhaust line in the far left of the diagram. Even if overflowing the inlet with N2, depending on the flow characteristics and pressures I could see a scenario where pump exhaust would flow out the inlet past your sampling point. Perhaps something is missing or I am interpreting this diagram incorrectly.

When overflowing with line with $N_2$ a valve is closed and the connection to the exhaust line is closed. We have added this information to the caption.

When overflowing the inlet line, the valve to the exhaust line in closed. When sampling air, the valve to the $N_2$ bottle is closed.

Why do the authors believe the HNO3 to 62 appears nonlinear in the inset figures on Fig 2 and 8?

We are dealing with very small signals when either acetate anion or $O_3$ are absent. The signal is close to background levels and may be influenced by variations therein. The point of the expanded scale in the insets is just to show that the signal is very low and detection of $HNO_3$ very inefficient.

Figure 9 and 10. These figures are difficult to interpret because there is not legend given in the figure and the colors are reused for different species. In 9, I believe the label attitude should be black and without reading the caption the reader has no idea what the pink line is for example. Blue is used in 3 places in Fig 9 for three different things.

Figure 9 (now 10) has been redrawn with more legends and corrected colours. Figure 10 ($NO_2$ 11) has also been redrawn and the lines better defined.

Page 8, line 238 needs a space between the and IOx

Correction made

Page ,8, line 241, suggest adding the word "of" in front of 100 greater.

Correction made

Page 9, line 265, there is a close parenthesis missing.

Correction made

Page 12, line 371, calibrations should be singular

Correction made

---

## Author Comment (AC5)

**Referee 3**
We thank the referee for a positive assessment of our manuscript. The comments (in black), our response (in blue) and changes to the manuscript (in red) are listed below.

The paper reports an investigation of the influence of the presence of HNO3 on the detection of NO3- and N2O5 when using chemical ionization mass spectrometry with I- as the reactant ion. Investigations are reported for especially the effect of ozone and humidity (that may affect the reactions through direct association of water molecules to ions). Additionally, the effect of PAN or PAA, that can release acetate anions causing extra reactions with HNO3, are reported. Finally, examples of field data, where the significance of the HNO3 trace gas is important for data interpretations of I-CIMS data, is given. The reported investigations were motivated by observations from airborne measurements using an I-CIMS where the magnitude and variation of the signal at mass 62 (NO3-) seemed to contradict previous beliefs on the measurement sensitivity to trace amounts of HNO3.

The theme of systematic sources of inaccuracies in CIMS measurements is of high relevance in general to atmospheric measurements and the presented results give important information on this aspect for the particular case of HNO3 trace gases in I-CIMS. As such the paper is of high relevance.

The supplementary information is not available, and as I learned from the editors, this information was actively removed by the authors prior to the review process. The supplementary information is in fact heavily needed to critically address the content of the paper, in particular

- Page 2 - Fig. S1 – should illustrate the data that motivated the re-investigation of the sensitivity of the I-CIMS detection of N2O5 to presence of the HNO3 trace gas
- Page 4 - Fig. S2 – should show details of the experimental calibration
- Page 9 - Fig. S3 – should give more information on the data corresponding in fig. 7
- Page 12 - Fig. S4 + Fig. S5 – should show data from The CAFE-Europa flights

I consider the removal of the supplementary information, where essential information is given and to which reference in made throughout the manuscript, as rather unserious and problematic. On this account, I would strongly hesitate to consider the paper for acceptance.

We posted the SI in reponse to this comment. As we state, the retraction of the SI was not intentional. If we had not intended to show the SI we would not have cited it in the manuscript.

General comments to the paper.
While the introduction is well written and easily accessible, the sections "experimental details" and "Laboratory characterization" could well be work over again for logic argumentation and clarity in the presentation.
I have the following specific comments, if the editors at all find the paper relevant for publication after the removal of the supplementary information.

We posted the SI in reponse to this comment.

Comment 1: Title and Abstract (e.g. line 11,14). The meaning of PAN is not given. Should be mentioned the same way as PAA.
In line 14 either use PAN and PAA or give the chemical formulas for both – a least be consistent.

In the atmospheric chemistry community the acronym PAN is more readily recognised than its correct, full name (peroxy acetyl nitric anhydride) so we prefer to keep it as PAN in the title. However, we have now defined PAN in the abstract:

....in the presence of peroxy acetyl nitric anhydride (PAN) or peroxyacetic acid (PAA).... and subsequently refer to the acronyms....

The loss of $CH_3CO_2^-$ via conversion to $NO_3^-$ in the presence of $HNO_3$ may represent a significant bias in I-CIMS measurements of PAN and PAA in which continuous cailibration (e.g. via addition of isotopically labelled PAN) is not carried out.

Comment 2: Page 2, Line 34-45 – several references to literature is missing . The statement "In very well known series of reactions …" must be backed up with a reference where one can find the reaction rate constants for these reactions.

We now list two citations:

In a very well-known series of reactions (Lightfoot et al., 1992; Atkinson et al., 2004), NO is oxidised (R1, R2) by reaction with $O_3$ or peroxyl radicals ($RO_2$) to $NO_2$

The statement on the "non-gas loss process" should also be backed by references

We now write:

Both $HNO_3$ and $N_2O_5$ have important, non-gas-phase loss processes (Crowley et al., 2010) such as uptake to particles and other surfaces.

The statement on rapid photolysis of NO3 should also be quantified with a reference and an actual number for the photolysis rate.

It is not possible to quote a single number for a photlysis rate without defining the solar zenith angle and cloud-cover. We prefer to state its approximate daytime lifetime and provide a citation.

The chain of reactions to form $N_2O_5$ is broken during the day as $NO_3$ is generally photolysed within a few seconds (Wayne et al., 1991)...

Comment 3: Page 3, line 81. I believe a rate of 380 s-1 would correspond to ~3 ms rather than ~2 ms as stated, or simply state 2.63 ms to keep the number of significant digits ?. I am also missing the reference that tells where the stated reaction rates comes from, i.e, where do the numbers 380 s^-1 and 1940 s^-1 come from.

As we quote only approximate temperatures, we prefer not to give an accurate lifetime w.r.t. thermal decomposition. We now provide citations and write:

At this inlet temperature and pressure, the lifetime of PAN with respect to thermal decomposition < 3 ms (IUPAC, 2021). For $N_2O_5$, the lifetime with respect to its thermal dissociation to $NO_2$ and $NO_3$ is ~0.5 ms (IUPAC, 2021)).

To appreciate the importance of these timescales, the authors should also specify the transport time of the various ions though the instruments sectors. I realize that some of this information can (partly) be reconstructed from the description of pages 6-7, but it should be clearly stated in this place, which would also ease the reading of the pages 6-7 a lot.

As this point of the manuscript we are dealing with the likelihood that PAN and $N_2O_5$ are thermally decomposed in the heated inlet. The residence time for PAN and $N_2O_5$ in the heated inlet is already stated on the preceding line of text. Adding text about resdience times in the IMR is not appropriate here.

Comment 4: page 3, line 86. Give reference to the origin of the stated reaction rate constants.

A citation (to IUPAC) has been added:

As NO reacts more rapidly with $NO_3$ than with $CH_3C(O)O_2$ at 170 °C ($k_{NO+NO3} = 2.3 \times 10^{-11}$ $cm^3$ molecule$^{-1}$ s$^{-1}$ , $k_{NO+CH3C(O)O2} = 1.4 \times 10^{-11}$ $cm^3$ molecule$^{-1}$ s$^{-1}$ (IUPAC, 2021))

What was the actual concentration of added NO ?

The concentration of NO is now listed:

....we periodically add NO (~ $5 \times 10^{12}$ molecule cm$^{-3}$) to the inlet to remove $CH_3C(O)O_2$ and thus eliminate sensitivity to PAN.

Comment 5: page 5, line 133. To appreciate that it is correct to normalize all signals to the primary ions signal, the reader need to be assured that the intensity of this peak (mass 127) is not affected (i.e. only marginally) by the reactions taking place. The authors needs to quantify this more precisely. I am in particular puzzled, since the [I(H2O)]/[I] ratio changes dramatically (approximately a factor of 7 (stated as 6, line 116)), so a least humidity most be important here

The loss of intensity at $m/z$ 127 is because water is present a much larger concentrations (percent in the boundary layer) than the trace-gases we try to detect (usually in the ppb range or lower). This is why the normalised signals are further corrected for relative humidity, as we describe. This is standard practice. We extend and clarify this:

Throughout the paper, when presenting raw data, we generally normalise the I-CIMS signal by dividing by the primary ion signal at $m/z$ 127. This is standard practise and corrects for drifts in the $CH_3I$ flow which may occur over several hours after the instrument was switched from standby mode to operational mode. It also accounts for longer term drifts caused by the weakening activity of the $^{210}Po$ source over the duration of a measurement campaign (months) or since the last calibration and for loss of detector sensitivity over similar time periods. We show later, that when adding large concentrations of reactants that significantly deplete the primary ion signal at $m/z$ 127 this procedure may lead to bias is some measurements. For this reason, when detecting $HNO_3$ at $m/z$ 190, we normalise to an interpolated signal at $m/z$ 127 that was measured when the air was scrubbed.

We have also added a new Figure (8) and text (new section 3.2) related to the loss of I- (and thus sensitivity to $HNO_3$) at $m/z$ 190 when adding large concentrations of $O_3$ to the IMR.

**Comment 6: page 5, line 133-135.** Could you expand on the explanation why reaction R6 (association of water to I-) is affected by the concentration of O3. This may be true in some indirect way, but to understand that, it really requires a more explicit explanation at this place in the paper.

Our apologies: The reference should have been to R7 and not to R6. We now write:

The weak signal in the absence of $O_3$ (blue data points) confirms the conclusions of previous studies that derive a low rate coefficient for reaction (R7)

Comment 7: page 5, line 146. The statement "This could be confirmed … m/z 62" seems to reflect an action that the authors speculate could prove their point that the first mentioned explanation for the sensitivity to O3 is not likely. Is this speculation or did you really do the suggested measurement ? It would be good to see the suggested evidence. (A figure in the missing supplementary information would be fine)

We use "could" not in the sense of "might have been able to" but as the past tense of "can" i.e. "were able to". To remove any ambiguity we now write:

This was confirmed by adding NO......

Comment 8: page 5, R10. On the right hand side, IO2- should be IO3-

Correction made

Comment 9: page 6-7 - discussion of the observed intensities of IOx-.

Given the rate coefficients of the various reactions, it seems straightforward to calculate the steady state distributions of I-, IO-, IO2-, and IO3- under the various conditions shown in figure 3. Following all the argument that ends on line 200, I believe it would worthwhile to do such a (simple) calculation and compare the result to the data in figure 3.

We considered this, but there are too many uncertainites including the reaction time, the greatly different rate coefficients presented in the two studies (Teiwes and Bhujel), the lack of information about the influence of water clusters, the potential impact of non-thermal reactions in these low-pressure studies and the possibility that we do not detect all IOx- anions with the same efficiency. The goal of the present study was not to ellucidate details of the reactions of IOx anions but to show that 1) they are formed when $O_3$ is present and 2) that they are likely candidates for the observed detection of $HNO_3$.

Moreover, following the discussion one page 7-8, in line 242, the statement is made that " confirming that the detection of IO3- in our experiment is inefficient" (see also line 223-24). I am puzzled if the efficiencies of the various IOx- components may affect the actual relative intensity ratios between them. The authors needs to clarify this issue.

We state that:

"However, the relative ion-abundance we observe at the detector does not necessarily reflect the relative concentration of the ions in the IMR and we cannot assign the individual contribution of any single $IO_X^-$ anion to $HNO_3$ detection."

This states clearly that we know neither the absolute numbers not the relative abundance of IOx- in the IMR. Given the uncertainties listed in the reply to the comment above, it is not obvious how we can further clarify this. We now weaken this statement by writing:

We note that the increase in signal at *m/z* 62 is about a factor of 100 greater than the reduction in the signal from $IO_X^-$, IMPLYING that the detection of $IO_3^-$ in our instrument is inefficient.

Comment 10: figure 7a and equation 2. Please explain the idea of suggesting the form in equation 2 to represent the data. I suppose it represents a short of saturation – but please clarify this more explicitly. Also the description in line 251 that "is clearly non-linear" is not really true: except for the lowest curve (18.5ppbv) all displayed curves are in fact linear to a good approximation as also suggested by eqn. 2, which is indeed almost linear at low [O3].

The amount of IOx anions formed depends exponentially on the rate coefficient (*k*), the concentration [$O_3$] of ozone in the IMR and the reaction time (t): $[IO_x]t = [I]_0(1-exp-k[O_3]t)$. This is why we chose this form of equation.

After equation 2 we have added:

Which reflects the expected exponential dependence of the concentration of $IO_x^-$ in the IMR on the $O_3$ concentration.